# Multiple Viral Infections Detected in *Phytophthora condilina* by Total and Small RNA Sequencing

**DOI:** 10.3390/v13040620

**Published:** 2021-04-04

**Authors:** Leticia Botella, Thomas Jung

**Affiliations:** 1Phytophthora Research Centre, Department of Forest Protection and Wildlife Management, Faculty of Forestry and Wood Technology, Mendel University in Brno, Zemědělská 1, 61300 Brno, Czech Republic; thomas.jung@mendelu.cz; 2Biotechnological Centre, Faculty of Agriculture, University of South Bohemia, Na Sadkach 1780, 37005 Ceske Budejovice, Czech Republic

**Keywords:** oomycete, dsRNA, *Bunyavirales*, *Totiviridae*, RdRp, mycovirus, estuaries

## Abstract

Marine oomycetes have recently been shown to be concurrently infected by (−)ssRNA viruses of the order *Bunyavirales*. In this work, even higher virus variability was found in a single isolate of *Phytophthora condilina*, a recently described member of *Phytophthora* phylogenetic Clade 6a, which was isolated from brackish estuarine waters in southern Portugal. Using total and small RNA-seq the full RdRp of 13 different potential novel bunya-like viruses and two complete toti-like viruses were detected. All these viruses were successfully confirmed by reverse transcription polymerase chain reaction (RT-PCR) using total RNA as template, but complementarily one of the toti-like and five of the bunya-like viruses were confirmed when dsRNA was purified for RT-PCR. In our study, total RNA-seq was by far more efficient for de novo assembling of the virus sequencing but small RNA-seq showed higher read numbers for most viruses. Two main populations of small RNAs (21 nts and 25 nts-long) were identified, which were in accordance with other *Phytophthora* species. To the best of our knowledge, this is the first study using small RNA sequencing to identify viruses in *Phytophthora* spp.

## 1. Introduction

*Phytophthora condilina* T.I. Burgess, sp. nov. belongs to *Phytophthora* phylogenetic Clade 6a, recently described in Western Australia from rhizosphere soil of dying *Casuarina obesa*, commonly known as swamp She-oak [1]. This tree is widespread in southern Western Australia, where it is commonly planted for agroforestry, particularly in salt-affected areas due to its highly tolerance to saline waterlogged conditions [2]. *P. condilina* is a sister species to *Phytopthora inundata* and *Phytophthora humicola* sharing a common ancestor with *Phytophthora balyanboodja*, *Phytophthora “personii”* and *Phytophthora gemini* [1]. Many species of Clade 6 are considered aquatic specialists [3], but *P. inundata* and *P. gemini* have a particular dominant aquatic lifestyle and are commonly found in brackish or even marine water. Although many Clade 6 species have been reported as pathogens, there are often other contributing environmental factors associated with the disease reports [1].

The first virus described from the genus *Phytophthora* was an alphaendornavirus found in an isolate of the undescribed *Phytophthora* taxon “douglas fir” in the USA [4]. Subsequently, similar virus strains were detected in *Phytophthora ramorum* isolates from several hosts in Europe [5]. The causal agent of potato late blight, *Phytophthora infestans*, has been more extensively studied and four RNA viruses were characterised. Phytophthora infestans RNA virus 1 and 2, (PiRV-1 and PiRV-2, respectively) seem to represent novel virus families [6,7], Phytophthora infestans RNA virus 3 (PiRV-3) is clustered with the newly proposed family “*Fusagraviridae*” [8] and Phytophthora infestans RNA virus 4 (PiRV-4) is an unclassified member of *Narnaviridae* [9]. PiRV2, which is 100% transmittable through asexual spores, has been demonstrated to enhance the sporangia production of *P. infestans*, thereby boosting the virulence of its host [10]. *Phytophthora cactorum* RNA virus 1 (PcRV1), a toti-like virus, was recently reported in an isolate of the pathogen *Phytophthora cactorum* from silver birch [11]. Furthermore, two novel endornaviruses were found co-infecting isolates of an unidentified *Phytophthora* species causing asparagus rot in Japan [12]. In other oomycetes, several (+)ssRNA viruses have been described in obligate biotrophic downy mildews including *Plasmopara halstedii* and *Plasmopara viticola* [13,14,15,16,17]. Virus-like particles and/or dsRNA have been observed in *Pythium irregular* [18], an unclassified gammapartitivirus was reported in *Pythium nuun* [19], a toti-like virus was also described from two strains of *Globisporangium splendens* (formerly *Pythium splendes*) and three virus-like sequences, Pythium polare RNA virus 1 (PpRV1), Pythium polare RNA virus 2 (PpRV2) and Pythium polare bunya-like RNA virus 1 (PpBRV1) were identified in *Pythium polare* infecting mosses in the Arctic [20]. Finally, multiple co-occurring bunyaviruses have also been described in marine oomycetes from the genus *Halophytophthora* [21].

Nowadays, a variety of next-generation sequencing (NGS) techniques are often applied to detect and describe fungal and oomycete viruses (mycoviruses). Stranded RNA sequencing appears to be the most predominant and successful technique but others like deep-sequencing small RNA (sRNA) libraries have also been proven to be efficient [22]. Since the RNA interference (iRNA) machinery targets possible detrimental non-self-nucleic acids, virus-infected host organisms are normally enriched with viral small interfering RNA (siRNA). Antiviral RNA silencing has been demonstrated to occur in different types of phytopathogenic fungi including the chestnut pathogen *Cryphonectria parasitica* [23,24], the plant pathogen *Rosellinia necatrix* [25], the mold fungus *Aspergillus fumigatus* [26], the arbuscular mycorrhizal fungus *Gigaspora margarita* [27]. Moreover, deep sRNA sequencing has been applied for de novo assembly of mycoviruses in the forest pathogen *Heterobasidion annosum* [28], the plant pathogen *Botrytis cinerea* [29] and a collection of marine fungi [30] but never in oomycetes. Nevertheless, sRNA-based pathways exist in *Phytophthora*, and Dicer-like (DCL), Argonaute (AGO) and RNA-dependent RNA polymerase (RDR) proteins have been identified in *P. infestans* [31,32]. Furthermore, two main sRNA populations with distinct regulation functions have been identified in *P. infestans*, *P. sojae*, *P. ramorum* and *P. parasitica* and *Haloperonospora arabidopsidis* [32,33,34].

Marine ecosystems appear to have an extraordinary virus abundance and diversity [35] with a range of biological and ecological implications, including control of microbial profusion and influence of global biogeochemical cycles [36,37,38,39]. However, our picture of the marine virosphere remains principally limited to bacteria, archaea and algae. In order to clearly understand the essential patterns and processes of virus evolution a wider range of organisms, including oomycetes, must be sampled and screened. Moreover, oomycetes are ubiquitous in marine, freshwater and terrestrial environments, and have a particularly interesting evolutionary history. Distantly related to brown algae they belong to the Kingdom Stramenopila (Heterokonta) and have been also associated with the nucleo-cytoplasmic large DNA viruses (NCLDVs) of the family *Megaviridae* in Tara Oceans [40].

Recently, multiple co-occurring bunya-like viruses have been described in marine oomycetes from the genus *Halophytophthora* isolated from brackish and marine water in saltmarshes in the South of Portugal [19]. In the same ecosystems several *Phytophthora* species have also been obtained including *P. condilina,* an oomycete with high tolerance to salinity and unclear phytopathogenic behavior recently described from aquatic and swampy ecosystems in Australia [1,2]. The aim of the present study was to increase knowledge about marine virus diversity by investigating the virome of a marine isolate of *P. condilina* from Portugal.

## 2. Materials and Methods

### 2.1. Provenance of the Virus Material

*Phytophthora condilina* isolate BD661 was collected in December 2015 from brackish water in a salt marsh in Parque Natural da Ria Formosa, (Olhão, Faro, Portugal) using an in situ baiting technique [41]. The isolate was identified using morphological characters and sequence analysis of the internal transcribed spacer (ITS1-5.8S-ITS2) region of the ribosomal DNA (ITS) and part of two mitochondrial loci [1,41]. The identification was confirmed by a NCBI BLASTn search (http://www.ncbi.nlm.nih.gov/BLAST/, last accessed on 31 March 2021) For the different purposes of the present study the isolate was grown in darkness on V8-juice Agar (V8A) [41] covered with cellophane (EJA08-100; Gel Company, Inc., San Francisco, CA, USA) for one to three weeks.

### 2.2. DsRNA and RNA Extractions

Double-stranded RNA (dsRNA) (Figure 1a) was extracted as previously explained [21]. DsRNA bands were cut and purified from the gel with Zymoclean Gel RNA Recovery Kit (Zimo research, Irvine, CA, USA).

Total RNA was purified from approximately 100 mg of fresh mycelium using RNAzol^®^ RT Column Kit Brochure, 2017, and treated with TURBO DNA-free™ Kit (Thermo Fisher Scientific, Waltham, MA, USA). RNA quantity was measured in Qubit^®^ 2.0 Fluorometer (Thermo Fisher Scientific, Waltham, MA, USA). The RNA quality was checked by Tape Station 4200 (Agilent, Santa Clara, CA, USA) resulting in a RNA integrity number (RIN) of at least 7.

### 2.3. Small and Total RNA Sequencing

#### 2.3.1. Small RNA (sRNA) Sequencing

Approximately 1 μg of total RNA eluted in RNase-free water was sent to Fasteris SA (Plan-les-Ouates, Switzerland). A polyacrylamide gel was prepared for visualization and selection of RNAs between 18 and 30 nucleotides. RNA library construction with Illumina TruSeq^®^ Small RNA Sample Preparation Kit (Illumina, San Diego, CA, USA). The library was sequenced in single reads (SE) (1 × 50 nt) run on an Illumina NextSeq 500 instrument.

#### 2.3.2. Total RNA Sequencing

Approximately 1 μg of total RNA eluted in RNase-free water was sent to MACROGEN (Seoul, Korea) for RNA library construction and deep sequencing. The library rRNA was depleted with Ribo-Zero rRNA removal kit (Human/Mouse/Rat) and built using the Illumina TruSeq^®^ Stranded RNA LT Sample Preparation Kit (Illumina, San Diego, CA, USA). The library underwent paired-end (PE) (2 × 101 nt) sequencing on a NovaSeq6000 (Illumina, San Diego, CA, USA).

The corresponding data are available in Sequence Read Archive (SRA) as BioProject PRJNA700086.

### 2.4. Bioinformatic Analyses

#### 2.4.1. Pipeline for Virus Assembling with Small RNAs

Low-quality reads were cleaned with TRIMMOMATIC 0.32. For the analyses, reads with sizes ranging 10 to 50 nt were selected and mapped to the genome sequence of *Phytophthora infestans* with BWA 0.7.5a. Consequently, the unmapped reads were extracted. BAM post-processing was performed with toolbox for manipulation of SAM/BAM files V. 1.13. De novo assembly was performed with VELVET 1.2.10.

#### 2.4.2. Pipeline for Virus Assembling with Total RNAs

Low quality reads were removed from the raw data (quality phred score cut-off: 20) as well as adapters and very short sequences (<25 bp) using Trim Galore (0.6.4_dev). The BWA mem program (v0.7.17-r1188) was applied for mapping samples against the viral and a host reference genome sequence (*Phytophthora parasitica*) with default settings. The count of mapped reads was extracted using SAMTOOLS 1.7. De novo assembly was performed using Trinity v2.5.1 program [42].

The final contig files from both RNA-Seq analyses were aligned to viral protein, non-redundant viral protein data bases using BLAST (v2.9.0+, BLASTn and BLASTX).

The numbers of reads of the final identified viral sequences were calculated with Bowtie2 v2.4.2 under the platform Geneious Prime^®^ 2020.0.4. For the calculation of the coverage depth we used the following formula: (Total reads (mapped to the final identified virus)) * (average read length)/virus genome or contig length. Average sRNA and total RNA read lengths were 30 and 101-long, respectively.

### 2.5. RNA Structures and Conserved Domains

DotKnot (https://dotknot.csse.uwa.edu.au/, last accessed on 26 February 2021) was used to search the H-type pseudoknots with maximum free energy (MFE). RNAfold web server (http://rna.tbi.univie.ac.at/, last accessed on 26 February 2021) was used to search the potential secondary structures with MFE in the 5′- and 3′-UTR. Predicted RNA secondary structures were visualized by Pseudoviewer (http://wilab.inha.ac.kr/pseudoviewer/, accessed on 26 February 2021). In order to search for conserved domains within the putative viral proteins the NCBI CD-search tool was used (https://www.ncbi.nlm.nih.gov/Structure/cdd/wrpsb.cgi, last accessed on 15 March 2021). IRESpy was applied in order to look for IRES in the UTRs of all the viruses (https://irespy.shinyapps.io/IRESpy/, accessed on 26 February 2021).

### 2.6. Genetic Variability

Pairwise identities of the nucleotide and amino acid sequences (Appendix A) were obtained after aligning the viral nucleotide and amino acid sequences by MUSCLE [43] and calculated by in Geneious Prime^®^ 2020.0.4.

### 2.7. Phylogenetic Trees

Maximum likelihood (ML) phylogenic trees were constructed using a rapid bootstrapping algorithm (Stamatakis 2008) in RAxML-HPC v.8 on XSEDE conducted in CIPRES Science Gateway [44]. Tree search was enabled under the GAMMA model to avoid thorough optimization of the best scoring ML tree at the end of the run. The Jones–Taylor–Thornton (JTT) model was chosen as substitution model for proteins. Bootstrapping was configured with the recommended parameters given by CIPRES Science Gateway. The resulting data were visualized using the software FIGTREE version 1.4.4. (http://tree.bio.edac.uk/softwa re/figtree/, last accessed on 16 March 2021).

### 2.8. Rapid Amplification of cDNA Ends (RACE) and Confirmation of Viruses’ Occurrence by Direct Reverse Transcription Polymerase Chain Reaction (RT-PCR)

In order to confirm the length of the putative totivirus-like genomes we used the SMARTer RACE 5′/3′ KIT (TAKARA BIO USA, Inc., Mountain View, CA, USA) as described in Botella et al. [21] and following the producer’s instructions.

The occurrence of each identified virus was confirmed by direct reverse transcription polymerase chain reaction (RT-PCR) with specific primers using both dsRNA (via cutting the bands) and total RNA as templates. A High-Capacity cDNA Reverse Transcription Kit (Applied Biosciences, Park Ave, NY, USA) was used for the cDNA synthesis. PCRs were performed with Hot Start *Taq* 2× Master Mix (New England BioLabs, Ipswich, MA, USA) including 25 µL Master Mix, 1 µL of each primer (10 mM) and 4 µL of cDNA in a total volume of 50 µL. All the primers used for the partial amplification of the RNA dependent RNA polymerase (RdRp) of each virus (Appendix A) were designed by Primer 3 2.3.7 under Geneious Prime^®^ 2020.0.4. Cycling conditions were set according to the manufacturer’s recommendations.

PCR products were visualised using gel electrophoresis (120 V; 60 min). Analysed fragments were separated on 1.5% agarose gel prepared with a TBE buffer (Merck KGaA, Gernsheim, Germany) and stained by Ethidium bromide (SIGMA-Aldrich, Steinheim, Germany). PCR products showing the amplicons of expected length were purified and sequenced by GATC BioTech (Eurofins; Konstanz, Germany) by both directions with the primers used for PCR amplification. Primer sequences can be consulted in Appendix A.

## 3. Results and Discussion

### 3.1. Phytophthora condilina Identification

Isolate BD661, baited from marine water in a saltmarsh in the south of Portugal, was homothallic with golden-brown oogonia, slightly wavy oogonial wall, thick-walled aplerotic oospores and both paragynous and amphigynous antheridia. Sporangia were non-papillate and proliferated internally in both a nested and extended way. This combination of morphological characters agreed with the description of *P. condilina* from Western Australia [1]. This identification was confirmed by a NCBI BLASTn search of the ITS (GenBank accession no. MW830150) and part of the mitochondrial *cox1* (MW836948) and *nadh1* (MW836949) genes of the isolate BD661.

### 3.2. Identification of Mycoviruses in P. condilina Isolate BD661

The dsRNA extraction from *P. condilina* isolate BD661 revealed the presence of one clear band usually around 6–7 kb (Figure 1a). Subsequently, sRNA and total RNA sequencing followed by de novo assembling and comparative BLAST searches identified 15 different putative viruses. Specific primers were designed for each viral sequence to confirm their presence in isolate BD661 by direct RT-PCR (Figure 1b).

A total of 48,096,027 reads (Appendix A) were generated from the sRNA library and 33,620,723 inserts of size 20 to 50 were selected for further analyses. When they were assembled against a related host genome (*P. infestans*) 16.60 *×* 10^6^ SE reads were unmapped and, selected for the following steps. De novo assembling obtained very short contigs, with an average length of 82 bases and the mapping in algae, plant and fungal virus databases failed to achieve decent computing coverage scores. Therefore, we decided to carry out total stranded RNA sequencing. The number of reads produced in the total RNA library (124,882,646) was almost three times higher than the small RNA reads with 94.2% of reads with a phred quality score of over 30 (Q30). When they were assembled against a related host reference sequence 1.12 × 10^6^ PE reads were unmapped and selected for the following steps. The final contig file had 6670 contigs with an average length of 395 nt, the maximum contig length was 9384 nt. Fifteen contigs were identified as possible viruses due to their significant E-values and identity percentage of their predicted amino acid sequences (Table 1). These results demonstrate that total RNA de novo assembling is considerably more efficient than sRNA de novo assembling, as previously reported [30]. Nevertheless, the mapping of sRNA-derived reads to the newly identified potential viruses (by total RNA-seq) showed very high read number and coverage depth (Table 1).

Thirteen contigs were similar to (-)ssRNA viruses, representing members of the order *Bunyavirales*. According to the BLASTX search in the non-redundant protein database (Table 1), they were mainly related to viruses hosted by fungal pathogens, such as *Botrytis cinerea* (Botrytis cinerea orthobunya-like virus 1, QKW91261), oomycetes, such as *Halophytophthora* sp. 04 (Halophytopthora RNA virus 1, MT277350) and the Arctic and Antarctic moss pathogen *Pythium polare* (Pythium polare bunya-like RNA virus 1, YP_09551341.1), and by arthropods like freshwater isopteran (Shahe bunya-like virus 1, KX884821). The pairwise (pw) sequence comparison (PASC) of the 13 bunyavirus-like nucleotide sequences showed an overall pw identity of 32.2% and always differed >10% between one another. As according to the ICTV there are not primary classification and delimitation criteria for genus and species in the order *Bunyavirales*, PASC and phylogenetic analyses seem to be the main point of reference to name new bunyaviruses. Therefore, we propose that these 13 bunyavirus-like sequences can be considered 13 putative different bunya-like viruses designated as Phytophthora condilina negative stranded RNA virus (PcoNSRV) 1–13 (Table 1), and we will avoid referring to them as different species. When the all nucleotide sequences were compared, the highest PASC values were observed between PcoNSRV5 and 8 (51.46%) and PcoNSRV6 and 9 (53.21%) (Appendix A), and between PcoNSRV5 and 8 (41.83%) and PcoNSRV6 and 9 (45.12%) when comparing the RdRp amino acid (aa) sequences (Appendix A). Conversely, the lowest values are perceived between PcoNSRV1 and 3 at nucleotide level (17.87%) and between PcoNSRV2 and 3 (6.20%) at protein level.

The BLASTX search of both totivirus-like sequences showed ~32% identity to unclassified viruses described in *Pythium polare* (Pythium polare RNA virus 1, YP_009552275) and in a diatom colony from rock pools in Tokyo Bay (Diatom colony associated dsRNA virus 17 genome type B, YP_009551504) (Table 1). The PASC values between the totivirus-like nucleotide sequences resulted in 42% identity (identical sites: 2880 nt). The alignment of both RdRp aa sequences showed a pw identity of 17% and the capsid protein (CP) only 14%. These data support the separation of these sequences into two different totivirus-like species because according to these virus species ICTV definition, less than 50% sequence identity at the protein level generally reflects a species difference. Totiviruses commonly reproduce stably within the cell as the cells grow. Different viral strains are expected to segregate relative to each other as the cells grow, whereas different virus species should be stably co-maintained [45]. Therefore, these two viruses were designated as Phytophthora condilina RNA virus (PcoRV) 1 and 2.

Bunya-like and toti-like viruses have been also identified in the same host isolate of *P. polare* [20] and the *Plasmopara viticola*-infected grapevine samples [16]. This may indicate that the co-infection of these unrelated viral genera is stable enough and they do not exclude each other. Indeed, multiple viral infections are known to occur in fungi and oomycetes in nature [16,17,20,46,47,48,49,50,51,52]. However, the virus-virus interactions and the limits of accumulation of multiple viruses in a single host remain unclear [53]. Initial multinfections are likely to be rare and ancient events which, once established, lead to long-term co-existence between virus and host in a non-lethal persistent life-style. Thus, fungal growth, which is characterized by cytoplasmic exchange in the vegetative and sexual phases, promotes the accumulation of multiple viruses which then develop coexistence strategies within an individual hypha, colony or hyphal network [48,54].

### 3.3. Genome Organization of New Putative Viruses Belonging to the Order Bunyavirales

Bunyaviruses are enveloped viruses with a genome consisting of three ssRNA segments (called L, M, and S). The S RNA encodes the nucleocapsid protein whereas the M glycoproteins and the L segments encode the RNA polymerase. Each genome segment is coated by the viral nucleoproteins (NPs) and the polymerase (L protein) to form a functional ribonucleoprotein (RNP) complex, necessary for the RNA replication and gene transcription [54]. However, in our study we have only discovered the L segment, which may be due to a lower copy number of the putative NP and other non-structural (Ns) associated proteins. Most likely, they are not conserved enough to be detected by homology, in contrast to what has been observed in other viruses including Penicillium roseopurpureum negative ssRNA virus 1 (PrNSRV1) [55] and Lentinula edodes negative-stranded RNA virus 2 (LeNSRV2) [56].

The L segment of the putative PcoNSRV1-13 encloses the complete coding region within the sequence using the standard genetic code, a single large open reading frame (ORF) coding for the RdRp (Figure 2). The RdRp nucleotide sequences range from ~6.3 kb (PcoNSRV3) to ~9.4 kb (PcoNSRV7) (Table 1). The rest of the sequence in 5*′* and 3*′* termini seems to constitute untranslated regions (UTRs), whose complete lengths remains unsure as RACE was not carried out in this case. Nevertheless, the size of the RdRp segment of PcoNSRV1-13 corresponds to the typical size of bunyavirus L segment.

Based on the aa sequence analysis, 11 bunyavirus-like sequences contained conserved regions belonging to pfam04196, Bunyavirus RdRp, which is the only member of the superfamily cl20265 (Figure 3a). Thus, PcoNSRV1 has a conserved region ranging from aa 680 to 1316 (expected e-value 8.21 × 10^−26^); PcoNSRV3 from aa 928 to 1204 (e-value 8.62 × 10^−4^); PcoNSRV4 from aa 1232 to 1637 (e-value: 4.46 × 10^−6^); PcoNSRV5 from aa 930 to 1446 (e-value: 4.79 × 10^−2^); PcoNSRV6 from aa 1363 to 1697 (e-value: 3.15 × 10^−7^); PcoNSRV7 from aa 1153 to 1788 (e-value: 8.14 × 10^−3^); PcoNSRV8 from aa 989 to 1467 (e-value: 1.26); PcoNSRV9 from aa 1063 to 687 (e-value: 7.77 × 10^−6^) and Bunya L protein N-terminus, an endonuclease domain found at the N-terminus of many bunyavirus L proteins (pfam15518) from aa 277 to 357 (e-value: 8.38 × 10^−4^); PcoNSRV10 from aa 731 to 1211 (e-value: 9.71 × 10^−8^); PcoNSRV11 from aa 1253 to 1661 (e-value: 3.14 × 10^−6^); PcoNSRV13 from aa 1331 to 1636 (e-value: 4.09 × 10^−4^). It was only in PcoNRV12 that we could not identify traditional conserved regions belonging to pfam04196. However, the amino acid alignment of the CDD regions (Figure 3a) show high similarities of the premotif A and motifs A (DxxxWx), B (XGxxNxxSS), C (SDD), D (KK) and E (ExxSx) with the rest of the bunyaviruses included in the alignment. Premotif A with the three basic residues inside (K, R and R/K) and, downstream, the glutamic acid (E), were also identified. The conserved aa triplet TPD (threonine), typical of bunyaviruses [57], was identified in PcoNSRV2 (position 221), and PcoNSRV4-13 (positions 336, 272, 346, 452, 319, 333, 88, 324, 92, 343, respectively) but it was SPD (Serine) in PcoNSRV1 (position 75) and only D was conserved in PcoNSRV3 (position 493). The doublet RY (arginine-tyrosine) was strictly conserved in PcoNSRV1 (position 680), and PcoNSRV3-13 (positions 614, 1064, 958, 1092, 1171, 1008, 1079, 760, 1048, 775, 1062) and partly conserved in PcoNSRV2 having K (Lysine) instead of Y (position 1123).

Interestingly, in PcoNSRV2, which contained a bunyavirus RdRp conserved domain from aa 1080 to 1603 (2.88 × 10^−8^), another conserved region was identified from aa 2563–2844 (9.98 × 10^−4^). This seems to belong to cas_TM1794_Cmr2 super family, a CRISPR (clustered regularly interspaced short palindromic repeat)–Cas (CRISPR-associated) protein Cas10/Cmr2, subtype III-B, typified by TM1794 from *Thermotoga maritima*, a marine hyperthermophilic anaerobic organism. The functional features of the proteins suggest that they comprise a specific repair system [58] and the ability to acquire immunity to viruses and plasmids [59]. The evolutionary implications of this result should be studied further.

### 3.4. Genome Organization of Two New Totivirus-Like Species

The complete genome sequences of PcoRV1 and PcoRV2 consist of a single segment of 5824 and 6240 bp (Figure 2, Table 1) with 58 and 55% G + C content, respectively.

Both PcoRV1 and 2 genomes contain two ORFs apparently encoding CP (ORF1) and RdRp (ORF2). In PcoRV1 both ORFs overlap (Figure 2) by seven nucleotides “CUGGUAG” (nt positions 3209–3215), which include the stop codon for ORF 2 (UAG) and the start codon for ORF1 (CUG, Leucine). This is a near-cognate codon that can, under certain circumstances, be also recognized by Metionine-tRNAi and is known to be the most efficient non-AUG initiation codon in many systems [60]. A heptanucleotide slippery site or shifty heptamer motif was observed at the end of the ORF1 (nt positions 3203–3209) with the general sequence X XXY YYZ (spaced triplets represent preframeshift codons) [61], where X represents A/G/C/U, Y represents A/U and Z represents A/C/U (GGAUUUC), which may facilitate −1 programmed ribosomal frameshifting in PcoRV1 transcripts (Figure 2). From the slippery site a short spacer region of 10 nt appears to precede a H-type pseudoknot between nt 3219–3246 (Figure 2), upstreams of the AUG, a key motif for translation of the downstream ORF, found at nt positions 3305 of PcoRV1.

In PcoRV2, ORF 1 and 2 overlap by 292 nts (from 3440 to 3731). The stop codon for ORF 2 is UAA and the putative start codon for ORF1 might be AUC (isoleucine), an alternative start codon shown to allow less efficient but still appreciable levels of initiation (2–30%) [60] AUC-initiation may have an inevitable consequence, a large proportion of ribosomes might scan past it and initiate instead at downstream the motif AUG (nts 3864–3867). Furthermore, we found the three elements needed to accomplish –1 ribosomal frameshifting in totiviruses and other RNA viruses within the overlapping ORF1-ORF2 region. The potential shifty heptamer motif (GGAUUUU) in nts 3506–3512, a pseudoknot downstream the shifty heptamer at nts 3515–3546 (Figure 2), and in between them, a very short spacer region of 3 nts (Figure 2).

Both PcoRV1 and 2 follow a non-canonical translation as seen in many other RNA viruses [62]. Alternative initiation (non-AUG codons) has been reported in many plant viruses but also in fungi and oomycetes. Phytophthora cactorum RNA virus 1 (PcRV1), Pythium polare RNA virus 1 (PpRV1), Ophiostoma minus totivirus (OmV) start RdRp-ORF with Leucine codons [11,20,46]. Non-canonical elongation (ribosomal frameshifting) is typical in fungal viruses of the families *Hypoviridae*, *Megabinaviridae*, *Fusagraviridae* and *Totiviridae*. This mechanism allows the virus to produce a demarcated ratio of Gag, Gag-Pol that is likely to be optimized for virion assembly and facile targeting of the replicative enzymes to the virion core. It also avoids the need to produce a separate mRNA for expression of the viral polymerase [62].

Based on the amino acid (aa) sequence analysis, conserved regions of the RT_like super family cl02808 and, in particular pfam02123, were identified in the putative RdRp sequence of PcoRV1 in the interval 310–539 (e-value: 4.33 × 10^9^) and in aa 270–655 (e-value 1.02 × 10^−3^) of PcoRV2. Conserved aa sequence motifs (I-VIII) of RdRp of PcoRV1 and 2 were aligned together with other totiviruses showing high similarity (Figure 3b). As the genome organization of PcoRV1 and 2 is similar to that found in members of the family *Totiviridae* [45] we propose that ORF1 (5′-proximal) encodes the CP (Gag) and ORF2 (3′-proximal) encodes the RdRp. In addition, PcoRV1 genome has a 1033-nt-long UTR at the 5′ terminus and a 111-nt UTR at the 3′ terminus (Figure 2). PcoRV2 genome has a 140-nt-long UTR at the 5′ terminus and a 50-nt UTR at the 3′ terminus. No IRES were identified in these regions but secondary structures were predicted in all of them with the exception of the PcoRV2- 5′ terminus (Data not shown).

### 3.5. Reverse Transcription Polymerase Chain Reaction (RT-PCR) Validation and Virus Depth of Coverage

The presence of all the putative viruses described in BD661 was confirmed using total RNA as template (Figure 1b). Surprisingly, presence of five of them (PcoNSRV1, 2, 3, 4 and 11) was also confirmed with dsRNA as template (Appendix A). The genome replication strategy of bunyaviruses and other (-)ssRNA viruses prevents the possibility of visualizing them by dsRNA extraction [63] because they require an encapsidation of their genomic and anti-genomic RNA by complexing them with viral nucleoprotein during synthesis. Apparently, this averts the development of long dsRNA sections by blocking complementary base pairing of the genomic and anti-genomic RNA [64]. This allows the virus to elude induction of cellular antiviral responses such as RNAi and interferon signaling, of which dsRNA is an effective activator [64,65,66]. In our study, we visualized just a very clear dsRNA band, which presumably belongs to the totivirus PcoRV1 but we lacked consistency in all the dsRNA extractions performed (Figure 1a) because this band (usually 6–7 kb) was slightly bigger in other extractions. On other occasions apart of the 6-kb sharp band there were other blurter bigger bands around 9–10 kb (average size of the bunyaviruses in BD661). Therefore, we decided to purify a piece of agarose from 6 to 20 kb, and go ahead with the protocol for RT-PCR detection.

PcoNSRV1, 2, 3, 4 and 11 were detected with sRNA-seq showing higher coverage depth than with total RNA-seq. Read number and coverage obtained by sRNA-seq was particularly high with PcoNSRV1, 2, 3 and 4. When viral dsRNA is detected it should mean that iRNA activates efficiently, which would explain the high depth of coverage of these viruses. However, PcoNSRV11, which showed lower coverage depth, was also detected by RT-PCR carried out from dsRNA. Therefore, we cannot define a clear pattern or any relation between dsRNA detection and levels of sRNA. Nevertheless, our results are in accordance with Donaire et al. [29], who detected Botrytis cinerea negative-stranded RNA virus 1 by sRNA-seq, and underline the importance of iRNA as an antiviral host-defense mechanism to negative stranded RNA viruses in oomycetes, as previously demonstrated in insects [66].

PcoRV1 had by far the highest read number in both RNA-seq techniques, being extremely high with sRNA-seq (Table 1). These results suggest that *P. condilina* iRNA mechanism is very effective against this dsRNA virus, as shown with other members of the family *Totiviridae* including *Rosellinia necatrix victorivirus* 1-W1029 [67]. This is in agreement with the statement that dsRNA produced during the replication is an activator of antiviral defenses and triggers iRNA in eukaryote organisms [64,65]. In contrast, PcoRV2 had the lowest values with both techniques and was not detected by RT-PCR using dsRNA as template, suggesting the possibility that this virus is less accumulated or is unstably replicating in *P. condilina* isolate BD661.

The graphical representation of the sRNA inserts (Appendix A) shows two predominant clusters of 21 and 25-nts. Accordingly, endogenous *s*RNAs characterised in *P. infestans*, *P. sojae*, *P. ramorum* and *P. parasitica* and *Haloperonospora arabidopsidis* have similar main sizes [31,32,33,34]. The 25-nt small RNAs seem to mainly derive from loci encoding transposable elements and are proposed to define a pathway of siRNAs silencing repetitive genetic elements. Further, the 21-nt small RNAs were primarily derived from inverted repeats, including a novel conserved microRNA family, several gene families, and Crinkler effectors [31]. No functional significance of microRNA-guided regulation of viruses has been reported so far but our work offers a framework for testing their role in *Phytophthora*.

### 3.6. Phylogenetic Analyses

A comprehensive reconstruction of the phylogenetic relationships of *P. condilina* bunya-like viruses with other (-)ssRNA viruses deposited in the GenBank shows that they cluster with unclassified viruses with genomic affinities of the order *Bunyavirales* (Figure 4) and distanced from members of the order *Mononegavirales* and *Jinchuvirales*. Thus, PcoNSRV 4–13 group with bunyaviruses mostly found in fungi and oomycetes suggested to be part of an unofficial family called Deltamycobunyaviridae [68]. Interestingly, many of the viruses of this cluster were previously reported in three isolates of *Halophytophthora* sp. 04 collected like *P. condilina* isolate BD661 from estuarine waters in the Portuguese Parque Natural da Ria Formosa but from a different site close to Faro [21]. This result confirms the high bunya-like virus abundance in oomycetes living in these estuarine ecosystems.

Separated clusters contain PcoNSRV1 and PcoNSRV2. PcoNSRV1 groups with Wuhan insect virus 3 (AJG39263) detected in *Aselus* sp. in China and *Halophytophthora* RNA virus 8 (MT277357), identified in three isolates of *Halophytophthora* sp. 04 in Santa Luzia, Portugal [21], previously designated as members of a novel unofficial family Epsilonmycobunyaviridae [21]. However, this name was previously assigned to a separated cluster in Nerva et al. 2019, so this possible virus family (including PcoNSRV1 and HRV8) should be named different. As the order *Bunyavirales* does not have a clear criterion to define species, genus and family we have decided not to use these unofficial family names in the tree. Finally, PcoNSRV2 clusters with two unclassified fungal orthobunyaviruses, Botrytis cinerea orthobunya-like virus 1 (QKW91261) and Tulasnella bunyavirales-like 1 (QPB44676). The latter has an unprecedented 12 kb genome organization containing three ORFs, the largest one encoding the RdRp, the second largest ORF encoding an unknown protein with Tom22 motifs (mitochondrial translocase family) and the smallest ORF encoding a nucleocapsid protein [70]. This unique genome feature remains to be confirmed in PcoNSRV2 because no 5′ and 3′ RACE has been carried out yet. Nevertheless, as pointed out in Section 3.1 PcoNSRV2 also has an interesting genomic feature as it contains a conserved region belonging to a CRISPR-associated protein Cas10/Cmr2, subtype III-B. A new taxon should be assigned to these viruses as they do not seem to belong to any of the existing ones. 

Likewise, PcoNSRV3 appears to belong to another novel taxon which also includes Shahe bunya-like virus 1 (APG79317), associated with small planktonic crustaceans (*Daphnia magna, D. carinata* and *Moina macrocopa*) from freshwater ecosystems in China [71] and bunya-like bridouvirus (QOW97246), detected in *Chlorarachnion reptans*, a macroalgae collected from marine sand in Victoria, Australia [72].

The reconstruction of the phylogenic relationships of P. condilina toti-like viruses with other related viruses from the GenBank clearly separated PcoRV1 and 2 in two different clades (Figure 5). PcoRV2 is included in clade I, which mainly contains mycoviruses belonging to the genera *Totivirus* and *Victorivirus*, but also members of *Thrichomonasvirus* and *Leishmaniavirus*. PcoRV2 appears closely related to toti-like viruses found in diatoms, aquatic organisms phylogenetically closely related to oomycetes [73]. Clade II includes PcoRV1, which clusters with other toti-like viruses infecting oomycetes, such as *P. cactorum*, *Plasmopara viticola*, *P. polare* and *Pythium splendens* (recently renamed *Globisporangium splendens*), and a number of toti-like viruses found in arthropods, many of them riparian and coastal species [71].

Both phylogenetic trees revealed close relatedness between viruses hosted by distinct oomycetes and fungi dwelling unalike ecosystems. Also, the relation of *P. condilina* viruses with viruses found in diverse organisms from similar brackish water habitats (Figure 4 and Figure 5). Some of them, crustacean hosts known to feed primarily on decaying vegetation, small invertebrates, microscopic algae (diatoms) [74], and most likely, also fungi and oomycetes. Remarkably, the class Oomycota includes marine holocarpic pathogens of nematodes, algae, crustaceans and mollusks [75], thus some of the viruses discovered in invertebrates may have originated from their gut mycoflora and/or parasites. Shedding light on the virus diversity of different organisms sharing the same ecological niche may, therefore, have major consequences for understanding the development that has shaped long-term virus biodiversity and evolution

## Figures and Tables

**Figure 1 viruses-13-00620-f001:**
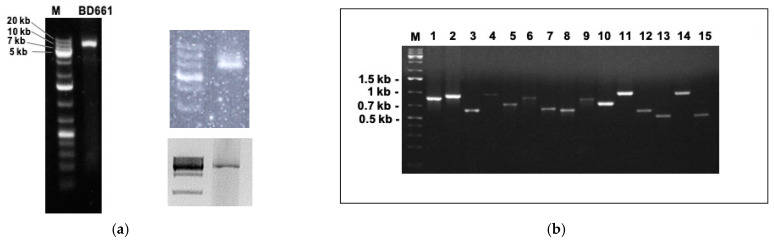
(**a**) dsRNA banding pattern of *Phytophthora condilina* isolate BD661 in three independent dsRNA extractions. DsRNAs from the second and third gel were purified in order to perform RT-PCR for virus detection. M, DNA marker (GeneRuler 1kb Plus DNA Ladder, 75–20,000 bp, Thermo Scientific). (**b**) RT-PCR amplicons using total RNA as template for cDNA synthesis, and amplified with specific primers for each virus. 1, PcoRV1; 2, PcoRV2; 3, PcoNSRV12; 4, PcoNSRV7; 5, PcoNSRV3; 6, PcoNSRV9; 7, PcoNSRV8; 8, PcoNSRV11; 9, PcoNSRV13; 10, PcoNSRV4; 11, PcoNSRV1; 12, PcoNSRV2; 13, PcoNSRV10; 14, PcoNSRV6; 15, PcoNSRV5. The specific size of each amplicon is shown in Appendix A.

**Figure 2 viruses-13-00620-f002:**
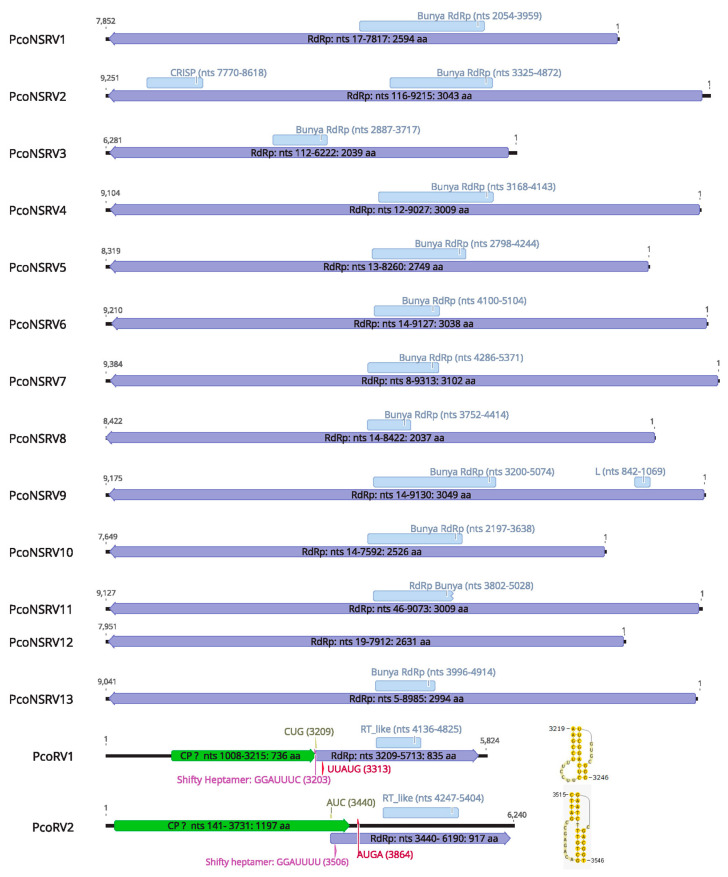
Genome organization of the bunya-like contig sequences (PcoNSRV1-13) and the full-length genome of the toti-like (PcoRV1-2) viruses described in this study. Violet and green boxes represent the open reading frame (ORF) detected and contain their information, size, genome region, protein encoded. Light blue boxes represent the conserved domains found in the CD-search. In PcoNSRV2, “CRISP” means CRISPR-associated protein Cas10/Cmr2 found by homology alignment. In PcoNSRV9, “L” means Bunya L protein N terminus; pfam04196. No conserved domains were detected in PcoNSRV3 and 12 by CD-search but by homology alignments. Pseudoknot in PcoRV1 with estimated free energy: −9.95 kcal/mol. Possible pseudoknot in PcoRV2 with an estimated free energy: −10.43 kcal/mol.

**Figure 3 viruses-13-00620-f003:**
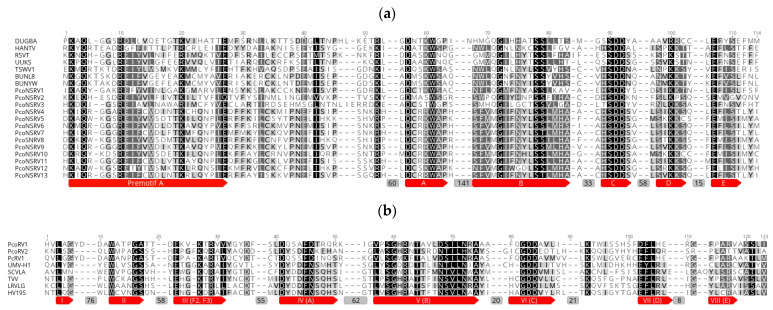
(**a**) Amino acid alignment showing conserved motifs A to E and premotif A within the RdRp of PcoNSRV1-13 and selected bunyaviruses. DUGBA, Dugbe virus (accession number Q66431); RSVT, Rice stripe virus (Q85431); UUKS, Uukuniemi *virus S23* (P33453); *TSWV*, Tomato spotted wilt virus (P28976); BUNYW*,* Bunyamwera virus (P20470); BUNL8, La Crosse virus L78 (Q8JPR2); HANTV*,* Hantaan virus 76–118 (P23456). Gray boxes with numbers represent the number of positions deleted in the MUSCLE alignment. (**b**) Conserved aa sequence motifs (I–VIII) of RdRp of totiviruses. PcRV1, Phytophthora cactorum RNA Virus 1 (QJS39952); *UMVH1*, *Ustilago maydis virus H1* (NP_620728); *SCVLA*, *Sacharomyces cerevisiae virus L-A* (Q87025); *TVV, Trichomonas vaginalis virus 3* (Q8V615); *LRVLG*, *Leishmania RNA virus 1* (Q02382); *HV19S*, *Heminthosporium victoriae virus 190S* (O57044).

**Figure 4 viruses-13-00620-f004:**
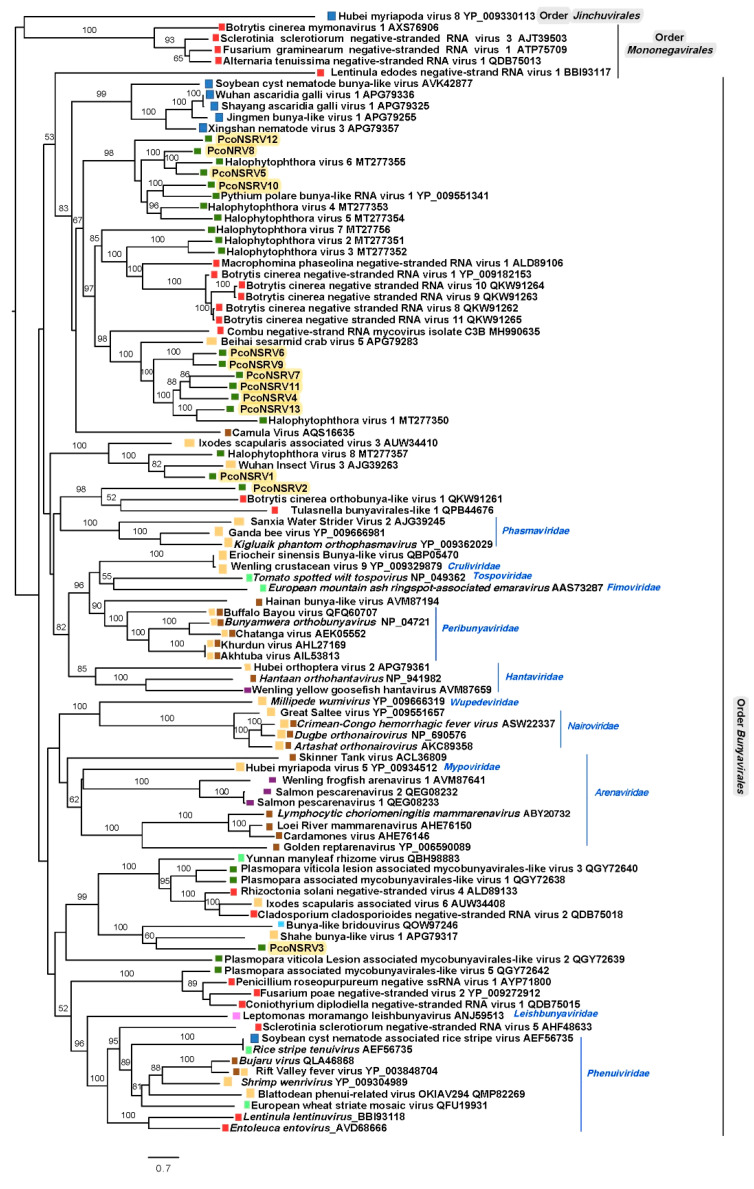
Maximum likelihood tree (RAxML) depicting the phylogenic relationship of the predicted RdRp of Phytophthora condilina bunya-like viruses with other complete RdRp belonging to related (-)ssRNA viruses from three different orders. Nodes are labelled with bootstrap support values ≥50%. Branch lengths are scaled to the expected underlying number of amino acid substitutions per site. Tree is rooted in the midpoint. Phytophthora condilina negative stranded RNA virus 1–13 are represented by their abbreviated names (PcoNSRV1-13). Family classification and the corresponding GenBank accession numbers are shown next to the virus names. Names in italics belong to the type species according to last update of the order *Bunyavirales* [69]. Colorful squares represent the virus host kingdom or phylum, 

 Fungi, 

 Nematoda, 

 Oomycota, 

 Arthropoda, 

 Plants, 

 Mammalia, 

 (Fishes) Chordata, 

 Ochrophyta (Heterokonta), 

 Excavata. Scale bar = 0.7 expected changes per site per branch.

**Figure 5 viruses-13-00620-f005:**
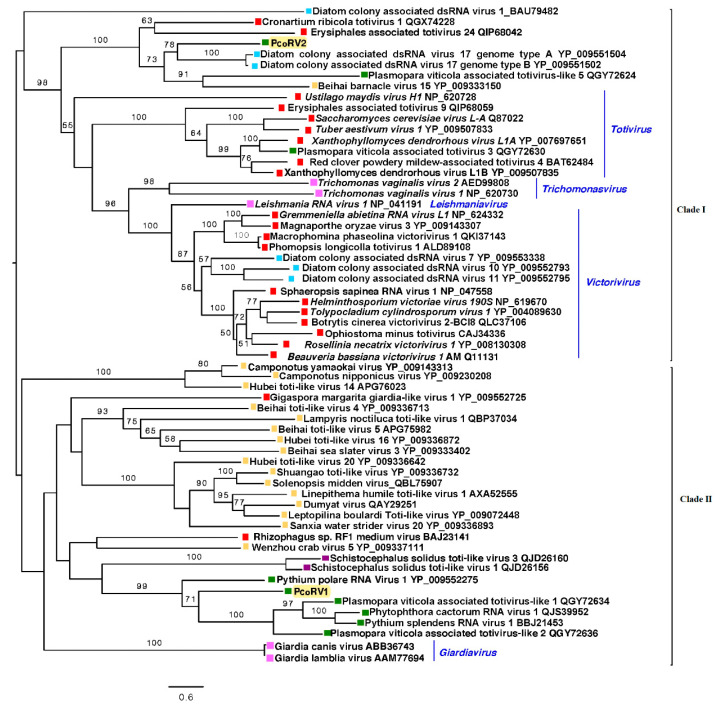
Phylogenetic analysis (RAxML) based on the predicted totivirus-like RdRp of *Phytophthora condilina* viruses with other complete classified and unclassified members of the family *Totiviridae*. Nodes are labelled with bootstrap support values ≥50%. Branch lengths are scaled to the expected underlying number of amino acid substitutions per site. Tree is rooted in the midpoint. Phytopthora condilina RNA virus 1 and 2 are abbreviated (PcoRV1, 2). Family classification and the corresponding pBLAST accession numbers are shown next to the virus names. Colorful squares represent the virus host kingdom or phylum, 

 Fungi, 

 Oomycota, 

 Arthropoda, 

 Chordata, 

 Ochrophyta (Heterokonta), 

 Excavata. Scale bar = 0.6 expected changes per site per branch.

**Table 1 viruses-13-00620-t001:** RNA dependent RNA polymerase (RdRp) sequences in GenBank most similar to the *Phytophthora condilina* viruses based on BLASTX search and parameters of their genome organization.

Acronym ^A^	Most smilar virus ^G^	E value	Q (%)	I (%)	L	Reads (Total RNA)	Cov. *	Reads (sRNA-Seq)	Cov. *	RT-PCR (Total RNA)	RT-PCR (dsRNA)
PcoRV1	Pythium polare RNA virus 1	9 × 10^−78^	39	32.18	5824	17,801	307	819,022	4212	YES	YES
PcoRV2	Diatom colony associated dsRNA virus 17 genome type B	5 × 10^−115^	38	33.87	6240	1873	30	12,279	59	YES	NO
PcoNSRV1	Halophytophthora RNA virus 8	0.0	81	78.97	7852	9447	103	354,982	1356	YES	YES
PcoNSRV2	Botrytis cinerea orthobunya-like virus 1	4 × 10^−42^	24	26.00	9251	10,793	117	178,909	580	YES	YES
PcoNSRV3	Shahe bunya-like virus 1	4 × 10^−140^	56	29.99	6281	11,269	181	177,627	848	YES	YES
PcoNSRV4	Halophytophthora RNA virus 1	0.0	76	39.06	9104	7435	82	95,833	316	YES	YES
PcoNSRV5	Halophytophthora RNA virus 6	0.0	67	60.31	8319	18,159	220	77,565	280	YES	NO
PcoNSRV6	Halophytophthora RNA virus 1	0.0	69	33.95	9210	8079	89	45,279	148	YES	NO
PcoNSRV7	Halophytophthora RNA virus 1	0.0	72	37.05	9384	10,979	118	41,068	131	YES	NO
PcoNSRV8	Halophytophthora RNA virus 6	0.0	68	48.31	8477	4925	59	33,109	117	YES	NO
PcoNSRV9	Halophytophthora RNA virus 1	0.0	71	33.35	9175	7905	87	23,860	78	YES	NO
PcoNSRV10	Pythium polare bunya-like RNA virus 1	0.0	84	47.84	7649	9119	120	15,956	59	YES	NO
PcoNSRV11	Halophytophthora RNA virus 1	0.0	76	36.06	9127	4225	47	7321	24	YES	YES
PcoNSRV12	Halophytophthora RNA virus 4	0.0	78	32.89	7951	7463	95	73,873	279	YES	NO
PcoNSRV13	Halophytophthora RNA virus 1	0.0	77	55.22	9041	5315	59	14,334	48	YES	NO

**^A^** Acronym of Phytophthora condilina RNA virus 1 and 2 (GenBank accession MW503714-15) and Phytophthora condilina negative stranded RNA virus 1–13 (MW503716-28), Q, Query cover, I, Identity, L, Virus sequence length, full-length genome sequence for PcoRV1 and 2 and contig length for PcoNSRV1-13, **^G^** Most similar viruses in GenBank (BLASTX), accession numbers: YP_00952275 (Pythium polare RNA virus 1), YP_009551504 (Diatom colony associated dsRNA virus 17 genome type B), MT277357 (Halophytophthora RNA virus 8), QKW91261 (Botrytis cinerea orthobunya-like virus 1), MT277350 (Halophytophthora RNA virus 1), MT277355 (Halophytophthora RNA virus 6), YP_009551341 (Pythium polare bunya-like RNA virus 1), MT277354 (Halophytophthora RNA virus 4). ***** Depth of coverage was calculated based on the formula: Total reads*read length/genome size (see materials and methods section).

## Data Availability

Data supporting reported results are available from GenBank under accession numbers MW503714-28. Total and small raw data are available in Sequence Read Archive (SRA) as BioProject PRJNA700086.

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
