# Peer review of "Multiple Viral Infections Detected in *Phytophthora condilina* by Total and Small RNA Sequencing"

_viruses, 2021, doi:10.3390/v13040620_

Round 1
Reviewer 1 Report
The authors have found and describe a novel bunyaviruses (13) and Toti-like viruses (2 complete genomes) in a single isolate of Phytophthora condilina isolated from estuarine water in Portugal. The work brings new information on the less-known virus diversity and coinfection ability in Oomycetes, and demonstrates the potential of small RNA vs. total RNA-seq to identify Phytophthora viruses.
Overall, the work is well executed and the manuscript is well written. However, below are some questions, comments and suggestions to be considered for improving the manuscript.
Introduction
L183 reverse transcription PCR
Results and discussion
L213 The band in figure 1a is much longer than 6 kb (over 10 kb). What does this dsRNA element correspond to?
L226-227 Please indicate the public database (and accession) to which the transcriptome data has been submitted
L228-234 which reference host genome was used in host depletion
L237 remove ‘Thus’
L237-238 was this a BLASTx against a virus database?
L243 Thirteen contigs
L253-254 You may also want to state that ICTV species delimitation criteria are not yet available for these groups of viruses, if that is the case, and explain the reasoning why these viruses are likely to represent distinct species.
L255-260 and related Tables S3, S4 Please clarify the lengths and regions of sequences used in the alignments (and how they relate to the species demarcation criteria in the order); the statements (and Tables) now seem a bit strange, since identities at the nucleotide level are much higher than at the protein level
L309-311 rephrase to something like “(UTRs), whose complete lengths remain unsure,…”
L314-320 and Figure 1a: It would be easier to follow if the conserved domain positions would be also indicated visually in virus genome structure images in Fig 1a (with a comment that pfam04196 in PcoNSRV3 and 12 were detected by homology alignments instead of conserved domain search). Same applied to the cas_TM1794_Cmr2 domain.
Also the reading frame in my opinion is not necessary, since the sequences do not represent the whole viral genomes.
L333 bunyavirus conserved region = Bunyavirus RdRp conserved domain ?
L342 consists remove s
L343 replace “and,” with “with” , and move word “respectively” to the end of sentence.
L344 ORFs
L372 “…Fusagraviridae and Totiviridae. This mechanism…”
L395-306 Maybe again indicate the positions of the conserved domains in the genome structure images (Fig.2a)
L434 and onwards
The raw iRNA and RNA-seq reads have been mapped to the contig sequences representing each of the novel virus genomes, and the depths of coverage with RNAseq and iRNA are reported. Did the mapping reveal any intraspecific variation (SNVs in the contig sequence), for example in relation to virus concentration in the host? Or were the sequences more or less homologous
L442-445 Please divide into to sentences, and rephrase slightly to explain more clearly what you mean.
L456 You mean PcRV2? Refer to Table 1 in this paragraph. Rephrase what you mean by “less present”.
L484-486
In Nerva et al. 2019 (Env Microbiol) The proposed “mycobunyaviridae” represent fungal viral families. Does the proposed “Epsilonmycobunyaviridae” (Botella et al 2020) contain also mycoviruses? If yes, please add them to the phylogenetic tree.
In addition, Nerva et al. have also proposed “Epsilonmycobunyaviridae” in Nerva et al. 2019 (Virus Res); the authors should point out the discrepancy and suggest alternative naming.
L509 toti-like viruses
Figure 2a. Could you indicate the locations of the pseudoknots (Fig 2b) in the genome structure image 2a? Indicate also the complete genome lengths of PcoRV1&2 in the images.
Indicate in the legend, that PcoRV1&2 are complete genomes, bunyaviruses are contig sequences
Figure 4. Indicate what “*“ means. Please address the discrepancy in Epsilonmycobunyaviridae (see previous comments). Correct spelling “Bunyavirales”, “Alphamycobunyaviridae”. How long was the aa sequence used in the phylogenetic analysis? Legend: change pBLAST to GenBank
Table 1. 1) The contig names are not necessary to state in the table. 2) Is “virus sequence length” actually the contig length? Or are the lengths of PcoRV1 and 2 the complete genome lengths? Please indicate.
Author Response
Introduction
L183 reverse transcription PCR. This change has been made
Results and discussion
L213 The band in figure 1a is much longer than 6 kb (over 10 kb). What does this dsRNA element correspond to? We lacked consistency in the extractions performed. Most of them had one sharp band around 6-7 kb and some blurter ones over it. In the first picture included the sharp band is bigger but we still think that corresponds to the totivirus 1. Nevertheless, if this picture is confusing, we can delete it. We decided to keep it because we wanted to show this lack of consistency in the dsRNA extractions.
L226-227 Please indicate the public database (and accession) to which the transcriptome data has been submitted. We have included this information in the section “materials and methods” and at the end of the manuscript, in the section “Data availability”
L228-234 which reference host genome was used in host depletion. We have included this information in this paragraph as suggested.
L237 remove ‘Thus’. This change has been made
L237-238 was this a BLASTx against a virus database? This information has been included
L243 Thirteen contigs. This change has been made
L253-254 You may also want to state that ICTV species delimitation criteria are not yet available for these groups of viruses, if that is the case, and explain the reasoning why these viruses are likely to represent distinct species. We agree with the reviewer comment and have changed the text to be more accurate in our statements.
L255-260 and related Tables S3, S4 Please clarify the lengths and regions of sequences used in the alignments (and how they relate to the species demarcation criteria in the order); the statements (and Tables) now seem a bit strange, since identities at the nucleotide level are much higher than at the protein level. We appreciate this comment and we have double checked the alignments and the tables. We re-aligned with MUSCLE the nucleotide and protein sequences and the distances remained the same. We have added the total number of sites of both alignments to the text.
L309-311 rephrase to something like “(UTRs), whose complete lengths remain unsure,…” This change has been made
L314-320 and Figure 1a: It would be easier to follow if the conserved domain positions would be also indicated visually in virus genome structure images in Fig 1a (with a comment that pfam04196 in PcoNSRV3 and 12 were detected by homology alignments instead of conserved domain search). Same applied to the cas_TM1794_Cmr2 domain. We appreciate this comment and have included this information in figure 2a.
Also the reading frame in my opinion is not necessary, since the sequences do not represent the whole viral genomes. We have removed this information from the ORF boxes.
L333 bunyavirus conserved region = Bunyavirus RdRp conserved domain? This change has been made as suggested.
L342 consists remove s. The sentence has been corrected
L343 replace “and,” with “with” , and move word “respectively” to the end of sentence. The sentence has been corrected
L344 ORFs. This mistake has been corrected
L372 “…Fusagraviridae and Totiviridae. This mechanism…” The sentence has been corrected
L395-306 Maybe again indicate the positions of the conserved domains in the genome structure images (Fig.2a). We appreciate this comment and have included this information in figure 2a.
L434 and onwards
The raw iRNA and RNA-seq reads have been mapped to the contig sequences representing each of the novel virus genomes, and the depths of coverage with RNAseq and iRNA are reported. Did the mapping reveal any intraspecific variation (SNVs in the contig sequence), for example in relation to virus concentration in the host? Or were the sequences more or less homologous. According to what our bioinformaticians suggested, the depth of coverage was calculated using the number of host-genome unmapped reads to the final contig file. And, there was no variation, the sequences were more and less the homologous.
L442-445 Please divide into two sentences, and rephrase slightly to explain more clearly what you mean. We have changed this paragraph in order to be clearer.
L456 You mean PcRV2? Refer to Table 1 in this paragraph. Rephrase what you mean by “less present”. We have changed less present for less accumulated.
L484-486
In Nerva et al. 2019 (Env Microbiol) The proposed “mycobunyaviridae” represent fungal viral families. Does the proposed “Epsilonmycobunyaviridae” (Botella et al 2020) contain also mycoviruses? If yes, please add them to the phylogenetic tree. In addition, Nerva et al. have also proposed “Epsilonmycobunyaviridae” in Nerva et al. 2019 (Virus Res); the authors should point out the discrepancy and suggest alternative naming.
We are very thankful for this comment because this is clearly a mistake in our previous article. We were not aware of the proposed Epsilonmycobunyaviridae in Nerva et al. 2019, which does not include PcoNSRV1 and HRV8. Nevertheless, as the order Bunyavirales does not have a clear criterion to define species, genera and families we have decided not to use these unofficial family names in our tree. None of these proposed families were included in Kuhn, J.H., Adkins, S., Alioto, D. et al. 2020 taxonomic update for phylum Negarnaviricota (Riboviria: Orthornavirae), including the large orders Bunyavirales and Mononegavirales. Arch Virol 165, 3023–3072 (2020). Besides, we consider that it can be confusing to use these proposed names here if ICTV name them differently later on. Anyway, we are open to discuss this point with the reviewer.
L509 toti-like viruses. This change was made
Figure 2a. Could you indicate the locations of the pseudoknots (Fig 2b) in the genome structure image 2a? Indicate also the complete genome lengths of PcoRV1&2 in the images. We have modified this figure to be clearer.
Indicate in the legend, that PcoRV1&2 are complete genomes, bunyaviruses are contig sequences. We have made this change
Figure 4. Indicate what “*“ means. Please address the discrepancy in Epsilonmycobunyaviridae (see previous comments). Correct spelling “Bunyavirales”, “Alphamycobunyaviridae”. How long was the aa sequence used in the phylogenetic analysis? Legend: change pBLAST to GenBank. We have corrected the tree accordingly
Table 1. 1) The contig names are not necessary to state in the table. 2) Is “virus sequence length” actually the contig length? Or are the lengths of PcoRV1 and 2 the complete genome lengths? Please indicate. We have completed the table as suggested.
Reviewer 2 Report
The manuscript presents the molecular characterization of thirteen negative-stranded RNA viruses ((-)ssRNA) and two dsRNA viruses associated with a single isolate of Phytophthora condilina. This work represents the first report of viruses associated to this oomycete. Additionally, the authors showed that the RNA silencing mechanism of P. condilina is acting against these mycoviruses.
Comments:
In the abstract, the authors should referred to the new (-)ssRNA viruses as bunya-like viruses, and no bunyaviruses. Totivirus-like viruses should be toti-like viruses.
In the introduction, authors should cite the paper Chiapello et al. 2020. Virus evolution 6(2) (in page 2 line 53), that described the virome associated to Plasmopara viticola. In the reference 27 (page 2 line 72) it was also showed that Botrytis RNA silencing machinery acts against all type of mycoviruses. Include also the reference of the sentence from line 92 to 95 (page 2).
In result section:
-Include the GenBank accession number of all viruses.
-Figure 1 should be Figure 3 and move to results section 3.5. Indicate what is in the pieces of gels beside dsRNA banding pattern of figure 1a
-Line 254, the authors claimed that they characterized 13 putative different species, but since there is no family or genus accepted by the ICTV to include these viruses, there is no specie demarcation criteria, then, at this point the authors should avoid to talk about 13 putative different species, and just talk about 13 different viruses. The situation is totally different in the case of the toti-like viruses.
-Line 275, include the reference Chiapello et al. 2020. Virus Evolution 6(2)
-Line 296, the sentence looks uncompleted
-Line 303, include the full name of PrNSRV1 and LeNRSV2
-Line 314, must be Figure 3a
-It is not clear what it is represented in the genomic scheme of toti-like viruses. AUG indicate the first start codon outside the overlapping region, if so, why there is no an alternative RdRp in PcoRV1? AUG motif in PcoRV2 is not in nt 3784, as has been indicated in line 361.
-Figures S4-S7 are no informative and could be eliminated
-Line 456, the authors mention that PcoRV1 was not detected by RT-PCR using dsRNA as template, but it is included in Figure S1.
-Review references, numbers 33 and 36 cite the same paper.
-Genus Entovirus was created for one new species, Ento-leuca entovirus, for Entoleuca phenui-like virus 1 (EnPLV-1), and genus Lentinuvirus was created for one new species, Lentinula lentinuvirus, for Lentinula edodes negative-strand RNA virus 2 (LeNSRV-2), both inside the family Phenuiviridae. This information is included in “Kuhn, J.H., Adkins, S., Alioto, D. et al. 2020 taxonomic update for phylum Negarnaviricota (Riboviria: Orthornavirae), including the large orders Bunyavirales and Mononegavirales. Arch Virol 165, 3023–3072 (2020).” Modify figure 4 according with the information included in the revision.
Author Response
Comments:
In the abstract, the authors should referred to the new (-)ssRNA viruses as bunya-like viruses, and no bunyaviruses. Totivirus-like viruses should be toti-like viruses. We have changed the text as suggested.
In the introduction, authors should cite the paper Chiapello et al. 2020. Virus evolution 6(2) (in page 2 line 53), that described the virome associated to Plasmopara viticola. In the reference 27 (page 2 line 72) it was also showed that Botrytis RNA silencing machinery acts against all type of mycoviruses. Include also the reference of the sentence from line 92 to 95 (page 2). We have included this reference and the suggested changes.
In result section:
-Include the GenBank accession number of all viruses. We included the accession numbers in the caption of the table 1 and they can also be found at the end of the paper (Data Availability Statement)
-Figure 1 should be Figure 3 and move to results section 3.5. Indicate what is in the pieces of gels beside dsRNA banding pattern of figure 1a. If the reviewer agrees we would prefer to leave the order of the figures as they are now and keep all the gel pictures together.
-Line 254, the authors claimed that they characterized 13 putative different species, but since there is no family or genus accepted by the ICTV to include these viruses, there is no species demarcation criteria, then, at this point the authors should avoid to talk about 13 putative different species, and just talk about 13 different viruses. The situation is totally different in the case of the toti-like viruses. We agree with the reviewer comment and have changed the text to be more accurate in our statements.
-Line 275, include the reference Chiapello et al. 2020. Virus Evolution 6(2). We have included this reference as suggested.
-Line 296, the sentence looks uncompleted. The authors are not sure of which sentence in particular but if the reviewer specifies it we are open to change it.
-Line 303, include the full name of PrNSRV1 and LeNRSV2. We have included these names as suggested.
-Line 314, must be Figure 3a. We have amended this mistake.
-It is not clear what it is represented in the genomic scheme of toti-like viruses. AUG indicate the first start codon outside the overlapping region, if so, why there is no an alternative RdRp in PcoRV1? AUG motif in PcoRV2 is not in nt 3784, as has been indicated in line 361. We have modified the figure, we hope it is clearer now.
-Figures S4-S7 are no informative and could be eliminated. We have eliminated these figures as suggested.
-Line 456, the authors mention that PcoRV1 was not detected by RT-PCR using dsRNA as template, but it is included in Figure S1. We apologize for this mistake, PcoRV1 was detected using dsRNA as template as well. We have corrected the sentence where this mistake appears.
-Review references, numbers 33 and 36 cite the same paper. We have amended this error and included a new reference with number 36 and 37.
-Genus Entovirus was created for one new species, Entoleuca entovirus, for Entoleuca phenui-like virus 1 (EnPLV-1), and genus Lentinuvirus was created for one new species, Lentinula lentinuvirus, for Lentinula edodes negative-strand RNA virus 2 (LeNSRV-2), both inside the family Phenuiviridae. This information is included in “Kuhn, J.H., Adkins, S., Alioto, D. et al. 2020 taxonomic update for phylum Negarnaviricota (Riboviria: Orthornavirae), including the large orders Bunyavirales and Mononegavirales. Arch Virol 165, 3023–3072 (2020).” Modify figure 4 according with the information included in the revision. We are thankful for this comment and have updated the tree according to this information.